# Comorbidities, medications, depression, and physical performance measures associated with severe cognitive impairments in community-dwelling adults

Gamar R. Almutairi[1], Noura R. Almegbas[1], Rawan M. Alosaimi[1], Maha A. Alqahtani[2], Saleh G. Batook[3], Ibrahim A. Alfageh[4], Mohammed M. Alshehri[5], Shuruq F. Alanazi[6], Ahmed S. Alhowimel [ID][1], Bader A. Alqahtani[1], Norah A. Alhwoaimel[1], Aqeel M. Alenazi [ID][1] *

1 Department of Health and Rehabilitation Sciences, Prince Sattam Bin Abdulaziz University, Alkharj, Saudi Arabia, 2 Department of Research and Innovation, King Abdullah International Medical Research Centre, Riyadh, Saudi Arabia, 3 East Jeddah General Hospital, Western Region, Jeddah, Saudi Arabia, 4 Abu Arish General Hospital, Jazan, Saudi Arabia, 5 Department of Physical Therapy, College of Applied Medical Sciences, Jazan University, Jazan, Saudi Arabia, 6 Primary Health Care, Ministry of Health, Riyadh, Saudi Arabia

* aqeelalenazi.pt@gmail.com, aqeel.alanazi@psau.edu.sa

**Data Availability Statement:** Data cannot be shared publicly because of ethical restrictions; the data contain potentially identifying and sensitive

## Abstract

Cognitive impairment negatively impacts health, psychological, social, and economic domains. Cognitive impairment commonly affects physical functions in older adults, whereas these are deteriorated. However, the prevalence and associated factors of cognitive impairment among community-dwelling adults in Saudi Arabia have not been investigated yet. This study aimed to examine the prevalence of severe cognitive impairment and its associated factors in community-dwelling older adults in Saudi Arabia. This cross-sectional study involved adults aged ≥50 years. Demographic data and clinical data, including number of medications and body mass index (BMI), were collected. Cognitive impairment and depressive symptoms were measured using Arabic versions of the Montreal Cognitive Assessment (MoCA) and Patient Health Questionnaire-9 (PHQ-9), respectively. The participants were divided into severe cognitive impairment and mild cognitive impairment or normal cognitive function groups based on a score of <20 or ≥20, respectively, using the MoCA. Physical measures included dynamic gait index (DGI) scores, timed up-and-go (TUG), 5 times sit-to-stand (5XSST), functional reach test, and 6-minute walk test. A total of 206 participants (female: n = 96) were included. The prevalence of severe cognitive impairment in the community-dwelling older adults was 12.6%. The number of chronic conditions (odds ratio [OR]: 2.31, p<0.001), number of medications (OR: 1.36, p = 0.003), and depressive symptoms using PHQ-9 (OR: 1.11, p = 0.009) were significantly associated with severe cognitive impairment after adjustment for other covariates, including age, sex, and BMI. Based on the physical function measures, only the scores for DGI (OR: 0.86, p = 0.003), TUG (OR: 1.16, p = 0.035) and 5XSST (OR: 1.25, p <0.001) were significantly associated with severe cognitive impairment. This study revealed a high prevalence of severe cognitive impairment among community-dwelling adults in Saudi Arabia. Its major risk

patient information that prevented us from sharing. Data are available from the Institutional Data Access / Ethics Committee of Sattam bin Abdulaziz University (contact via IRB-sciences@psau.net) for researchers who meet the criteria for access to confidential data.

**Funding:** This study was supported via funding from Prince Sattam Bin Abdulaziz University project number (PSAU/2024/R/1445). The funders had no role in study design, data collection and analysis, decision to publish, or preparation of the manuscript.

**Competing interests:** The authors have declared that no competing interests exist.

factors include depressive symptoms, number of chronic conditions and medications, and physical measures, including DGI, TUG, and 5XSST.

## Introduction

The life expectancy in Saudi Arabia has increased over several recent decades, resulting in a rapid increase in the proportion of older people. The General Authority for Statistics in Saudi Arabia estimated the percentage of persons aged ≥65 years accounts for 2.4% of the total population in 2019 [1]. By 2050, the elderly are predicted to account for 18.4% of the total population [2]. Consequently, age-related conditions require close monitoring and continuous care. Aging is associated with functional deterioration affecting various body systems, including musculoskeletal, cardiovascular, and cognition systems [3–6]. Cognitive decline is more prevalent with age, affecting well-being and bodily functions necessary to maintain total independence, self-care, and autonomy [7, 8].

Cognitive impairment is defined as a state of impaired cognition involving different domains, including memory, executive functions, attention, language, and visuospatial skills. The stages of cognitive impairment are very mild, mild, moderate, moderately severe, severe (middle dementia) and very severe (severe dementia). Mild cognitive impairment (MCI) is a stage of cognitive decline that is not sufficiently severe to require help with everyday tasks [9].

Several risk factors can be used to predict cognitive decline in elderly populations. Alkhunizan et al. states that age, educational level, smoking, overweight, diabetes, high blood pressure, and hyperlipidemia are related to MCI and dementia [10]. The prevalence of MCI was 15–20% in people aged ≥60 years, making it a frequent disorder encountered by physicians. However, MCI is a transitional stage to dementia[10, 11].

Cognitive decline was associated with depressive symptoms among Chinese community-dwelling older adults [12]. Furthermore, cognitive decline was associated with depressive symptoms in Saudi community-dwelling adults [13]. However, this study relate the depressive symptoms with severe cognitive impairments. Therefore, using depressive symptom cutoff scores with severe cognitive impairment is crucial for screening and to establish interventional approaches in this population.

Cognitive functioning is associated with physical functioning. Among older adults, physical functions, including gait, balance, and mobility, are affected; however, these functions are more frequently deteriorated in the elderly with cognitive impairment [14–17]. A cross-sectional study investigating the relationship between mental impairment and each component of activity in daily living (ADL) in elderly populations showed that cognitive status can be a predictive factor for ADL in the elderly [18].

Cognitive impairment can negatively impact health, psychological, social, and economic domains. Individuals with cognitive impairment have a poor quality of life, a high risk of falls, and an increased rate of disability and functional limitations [19]. Progression of cognitive impairment increases dependency levels, affecting the amount of caregiving required. In addition, people with cognitive impairment have an increased risk of death, a lower quality of life, poor quality of mental health, and longer hospital stays [8].

The World Health Organization (WHO) considers severe cognitive impairment to be the seventh-leading cause of death involving all diseases [20]. Caregivers spend an average of 6.7 h on informal care for people with dementia [21]. The annual costs of dementia are estimated at US$ 818 billion and are predicted to increase to US$ 2 trillion by 2030 [22]. Nevertheless, the

prevalence of and factors associated with severe cognitive impairment among community-dwelling adults in Saudi Arabia have not been investigated yet. Therefore, this study aimed to examine the prevalence of severe cognitive impairment and its associated factors in community-dwelling adults in Saudi Arabia. The findings of this study would provide a reliable basis for preventive community measures to decrease severe cognitive impairment among older adults in Saudi Arabia.

## Methods

### Study design and participants

This was a cross-sectional study performed in Saudi Arabia between February 2, 2022, and April 30, 2022, using convenience sampling. The inclusion criteria was as follows: age ≥50 years, being able to perform functional tests safely, being able to read and write in the Arabic language, and being a Saudi citizen. Non-Saudi adults, age <50 years, and an inability to read and write in Arabic, were considered as exclusion criteria since some outcome measures that were self-reported were validated in Arabic. Participants were recruited from different geographical regions across Saudi Arabia (Riyadh, Alkharj, Jazan, Jeddah, Arar, Hail, and Tabuk). The data was collected by trained physical therapists who recruited participants from different communities (mosques, malls, clinics, and other social-gathering locations). A total of 418 participants were approached, and 212 participants were included in the current study (inclusion rate, 50.7%). The reasons for refusal to participate were lack of time and transportation. The included participants signed written consent following the mandates of the Declaration of Helsinki. This study was approved by the Research Ethics Committee of the Prince Sattam Bin Abdulaziz University (No. RHPT/021/017)

### Demographics and anthropometric data

Data were collected using a standardized questionnaire and all the interviewers were well-trained to mitigate information bias [23]. Data collection commenced with demographic data, including age, gender, and body mass index (BMI). Age was recorded in years and categorized into six groups (i.e., 50–54, 55–59, 60–64, 65–69, 70–74, and > 74 years). Gender was reported as male or female. Weight and height were measured in kg and cm, respectively. BMI was calculated by dividing weight in kg by squared height in $m^2$ and categorized into three groups (i.e. normal weight [BMI < 25], overweight [25 < BMI <30] and obese [BMI ≥ 30]). Marital status was dichotomized as married or divorced/widowed.

### Outcome measures

Cognitive function was assessed using the Arabic version of the Montreal Cognitive Assessment (MoCA) [24]. The MoCA comprises 7 parts, including visuospatial (maximum score, 5/5), animal name identification (3/3), attention (6/6), language (3/3), abstraction (2/2), delayed recall (5/5), orientation (6/6), and education (1/1). The total score ranged from 0 to 31 with a higher score indicating better cognitive functions. Participants were classified into severe cognitive impairment and MCI/normal cognitive function groups based on a score of < 20 and ≥ 20, respectively [25–27].

### Exposure factors

Chronic diseases have been evaluated with different definitions and numbers in previous research. One of the definitions used chronic non-communicable diseases involved 10 conditions [28]. We used a similar definition with some revisions related to the prevalence of

diseases in Saudi Arabia. Major chronic diseases, including hypertension, diabetes, cardiovascular disease, cancer, pulmonary disease, dyslipidemia, neurological diseases, osteoporosis, arthritis, and low back pain, were recorded via self-reported diagnoses. The total number of chronic diseases and currently used medications were also recorded. Self-reported chronic diseases had adequate validity with good agreement to well-known chronic conditions, such as diabetes, hypertension, and cardiovascular diseases [29, 30]. The number of currently used medications was reviewed by the research team and then recorded.

Depressive symptoms were measured using the Arabic version of Patient Health Questionnaire 9 (PHQ-9), which is considered as valid and reliable [31, 32]. PHQ-9 includes nine self-reported questions related to depressive symptoms, including anhedonia, depressed mood, sleep troubles, fatigue, change in appetite, decreased self-esteem, concentration disturbance, psychomotor disorder, and suicidal thoughts [33]. The participants reported the impact of symptoms through questions, such as 'How difficult have these problems made it for you to do your work, take care of things at home or get along with other people?.' The answers employed a four-point scale (0–3): 'not at all = 0', 'several days = 1', 'more than half the days = 2,' and nearly every day = 3'. The total score ranges from 0 to 27, with greater scores indicating more depressive symptoms.

Gait and balance while walking was assessed using the Arabic version of the dynamic gait index (DGI), which was previously translated and validated [34–36]. DGI evaluates the participants' ability to walk steadily and modify their gait in response to challenging tasks. The participants' walk performance was evaluated via the following gait tasks: steady-state walking, walking with changing speeds (fast/comfortable/slow), walking with head turned up and down, walking while stepping around and over obstacles, spinning while walking, and stair climbing. The scores were based on a 4-point scale, as follows: '3 = no gait dysfunction', '2 = minimal impairment', '1 = moderate impairment,' and '0 = severe impairment'. The highest score possible was 24 points, indicating superior performance.

Timed up-and-go (TUG) was used to measure balance, mobility, ability to walk, and risk of fall [37]. The examiner tasked the participant to stand up, walk for 3 meters, turn around, walk back, and sit down on a chair. Greater scores in seconds indicated poorer performance in balance and mobility and a greater risk of falling.

Dynamic balance and stability were measured using the functional reach test (FRT) [38]. The participants were instructed to extend forward their dominant arm to the greatest extent possible. An examiner measured the distance between the starting and ending points of the third metacarpal bone in cm. A lower distance indicates poor balance and a high risk of falling.

Endurance and aerobic capacity were assessed using the 6-minute walk test (6-MWT), following a standardized protocol [39]. 6-MWT examines the distance walked by participants over a period of 6 minutes. A greater distance (in m) indicates better endurance and performance.

The 5 times sit-to-stand test (5XSST) was used to measure functional lower extremity strength [40]. 5XSST measures the time required to complete five repetitions of the sit-to-stand position on a chair without using the hands. The shorter the time to complete the test, the better the outcome.

## Statistical analysis

Descriptive statistics are used to define categorical variables and means, and standard deviations for continuous variables for each group. The two groups (severe cognitive impairment and MCI groups) were compared based on demographics and clinical variables using the Chi-

squared or Fisher's exact test for categorical variables and independent Student's *t*-test for continuous variables.

Multiple binary logistic regression, odds ratio (OR), and 95% confidence interval (95% CI) were used to assess the potential associations between the risk factors and physical function measures of severe cognitive impairment. The variables included in the regression model were selected based on their relevance and potential association with cognitive impairment. The following two models were developed based on adjustments: model 1 was unadjusted, and model 2 was adjusted for age, sex, and BMI to control the key factors that have been shown to influence cognitive outcome. Previous studies employed unadjusted models [10, 41]. Missing variables were handled using case-wise deletion.

Receiver operating characteristic (ROC) curves were constructed to determine the cut-off scores for significant risk factors for severe cognitive impairment. The area under the ROC curve (AUC) indicates the overall accuracy of the model in detecting the presence or absence of the outcome, such as severe cognitive impairment. The maximum AUC is the best cut-off score, measured by the largest Youden's index (sensitivity + [1 − specificity]). Sensitivity and specificity were calculated to determine true positives and true negatives, respectively. AUC values were interpreted according to an arbitrary guideline, no (AUC < 0.5), poor (0.5 ≤ AUC < 0.7), acceptable (0.7 ≤ AUC < 0.8), excellent (0.8 ≤ AUC < 0.9), and outstanding (AUC > 0.9) discriminant ability [42].

Finally, a sensitivity analysis was conducted using multiple imputation method to compare the results after five multiple imputations for missing observations using age, gender, and BMI as predicators with the results from the original dataset using case wise deletion. An alpha level of 0.05 was used for all analyses. All analyses were performed using IBM SPSS for Mac v.25.0 (SPSS Inc. Chicago, IL, USA).

## Results

Our final analysis included 206 participants. Table 1 presents the demographics and clinical characteristics of participants with and without severe cognitive impairment. The prevalence of severe cognitive impairment among community-dwelling adults was 12.6% (n = 26); this was higher in women than that in men and in the 60–64-year age compared with that in the other age categories. Most of the variables, including age, sex, number of chronic conditions, number of medications, and depressive symptoms, differed significantly between the severe cognitive impairment and MCI or normal groups, determined using the PHQ-9, DGI score, TUG, 5XSST, FRT, and 6MWT results.

Table 2 shows the results of multiple binary logistic regression examining the associated risk factors of severe cognitive impairments, with OR and 95% CI. The number of chronic conditions (OR: 2.31, 95% CI [1.57, 3.41], p < 0.001), number of medications (OR: 1.36, 95% CI [1.11, 1.67], p = 0.003). and depressive symptoms determined using the PHQ-9 (OR: 1.11, 95% CI [1.03, 1.21], p = 0.009) were significantly associated with severe cognitive impairment after adjustment for other covariates, including age, sex, and BMI. Physical function measures showed that the DGI (OR: 0.86, 95% CI [0.77, 0.94], p = 0.003), TUG (OR: 1.16, 95% CI [1.01, 1.34], p = 0.035) and 5XSST (OR: 1.25, 95% CI [1.11, 1.39], p < 0.001) scores were significantly associated with severe cognitive impairment after adjustment for other covariates, including age, sex, and BMI.

Table 3 summarizes the results of the ROC curve analysis, including AUC and identified cut-off scores for each risk factor (number of chronic conditions, number of medications, PHQ-9, DGI, TUG, 5XSST) associated with severe cognitive impairment, along with their sensitivity and specificity. Regarding the number of chronic conditions, having ≥ 2.5 chronic

**Table 1. Demographic, anthropometric and clinical factors of the participants.**

| Factors | Severe cognitive impairment (n = 26) | MCI/normal cognitive function (n = 180) | p-value* |
|---|---|---|---|
| Age category | | | 0.018 |
| 50–54 years, n (%) | 5 (19.2) | 54 (30.0) | |
| 55–59 years, n (%) | 4 (15.4) | 47 (26.1) | |
| 60–64 years, n (%) | 8 (30.8) | 50 (27.8) | |
| 65–69 years, n (%) | 2 (7.7) | 18 (10.0) | |
| 70–74 years, n (%) | 3 (11.5) | 4 (2.2) | |
| $^3$75 years, n (%) | 4 (15.4) | 7 (3.9) | |
| Sex male/female | 4/22 | 92/88 | 0.001 |
| Marital status | | | 0.003 |
| Married, n (%) | 17 (65.4) | 161 (89.4) | |
| Divorced/widowed, n (%) | 9 (34.6) | 19 (10.6) | |
| BMI category | 29.8±5 | 28.5±5 | 0.41 |
| Normal BMI ($<$ 25), n (%) | 4 (15.4) | 50 (27.8) | |
| Overweight (25–29.9), n (%) | 11 (42.3) | 66 (36.7) | |
| Obese ($^3$30), n (%) | 11 (42.3) | 64 (35.6) | |
| Number of chronic conditions (mean±SD) | 3.30±1.6 | 1.39±1.28 | $<$ 0.001 |
| Number of medications (mean±SD) | 2.96±2.7 | 1.22.66±1.6 | 0.004 |
| PHQ-9 (mean±SD) | 8.62±5.66 | 4.66±4.77 | $<$ 0.001 |
| DGI (mean±SD) | 16.12±6.18 | 20.40±3.77 | 0.002 |
| TUG, sec (mean±SD) | 12.58±3.4 | 10.52±2.9 | 0.001 |
| 5XSST, sec (mean±SD) | 19.32±8.7 | 13.44±3.47 | 0.002 |
| FRT, cm (mean±SD) | 24.23±6,2 | 27.10±8.16 | 0.047 |
| 6MWT, m (mean±SD) | 377±89 | 428±106 | 0.026 |

*Notes*: *p-value was based on Chi-square or Fisher's Exact test and independent t-test for categorical and continuous variables, respectively.

MCI: mild cognitive impairment, BMI: body mass index, PHQ-9: Patient Health Questionnaire-9, DGI: dynamic gait index, TUG: timed up and go, 5XSST: 5 times sit-to-stand, FRT: functional reach test, 6MWT: 6-minute walk test

conditions demonstrated an excellent discriminant ability in predicting severe cognitive impairment (Fig 1). A cut-off score of 2.5 for the number of medications had an acceptable ability to detect the presence of severe cognitive impairment in older adults (Fig 2). The cut-off

**Table 2. Binary logistic regression for severe cognitive impairment versus risk factors.**

| | Unadjusted model n = 206 | | Adjusted model n = 206 | |
|---|---|---|---|---|
| Factors | OR (95% CI) | p-value | OR (95% CI) | p-value |
| Number of chronic conditions | 2.35 (1.69, 3.28) | $<$ **0.001** | 2.31 (1.57, 3.41) | $<$ **0.001** |
| Number of medications | 1.44 (1.19, 1.73) | $<$ **0.001** | 1.36 (1.11, 1.67) | **0.003** |
| PHQ-9 | 1.13 (1.06, 1.22) | **0.001** | 1.11 (1.03, 1.21) | **0.009** |
| DGI | 0.83 (0.77, 0.90) | $<$ **0.001** | 0.86 (0.77, 0.94) | **0.003** |
| TUG | 1.20 (1.07, 1.36) | **0.003** | 1.16 (1.01, 1.34) | **0.035** |
| 5XSST | 1.24 (1.13, 1.38) | $<$ **0.001** | 1.25 (1.11, 1.39) | $<$ **0.001** |
| FRT | 0.96 (0.90, 1.01) | 0.09 | 0.94 (0.88, 1.01) | 0.08 |
| 6MWT | 0.99 (0.99, 0.99) | **0.02** | 0.99 (0.99, 1.00) | 0.27 |

*Notes*: The adjusted model included age, gender, and BMI.

OR: odds ratio, BMI: body mass index, PHQ-9: Patient Health Questionnaire-9, DGI: dynamic gait index, TUG: timed up and go, 5XSST: 5 times sit-to-stand, FRT: functional reach test, 6MWT: 6-minute walk test

**Table 3. ROC curve and cut-off scores for the significant risk factors of severe cognitive impairment.**

| Variables | AUC (95% CI) | Cut-off score (sensitivity, specificity) |
|---|---|---|
| Number of chronic conditions | 0.82 (0.74, 0.91) | 2.5 (0.69, 0.80) |
| Number of medications | 0.71 (0.59, 0.82) | 2.5 (0.50, 0.83) |
| PHQ-9 | 0.73 (0.62, 0.83) | 6.5 (0.65, 0.74) |
| DGI | 0.73 (0.63, 0.83) | 21.5 (0.50, 0.89) |
| TUG | 0.69 (0.58, 0.79) | 11.77 (0.58, 0.74) |
| 5XSST | 0.77 (0.66, 0.87) | 15.60 (0.62, 0.82) |

PHQ-9: Patient Health Questionnaire-9, DGI: Dynamic Gait Index, TUG: Timed Up and Go, 5XSST: 5 times sit-to-stand.

*Notes*: AUC < 0.5 and 0.5 ≤ AUC < 0.7 denote no and acceptable discriminant ability, respectively

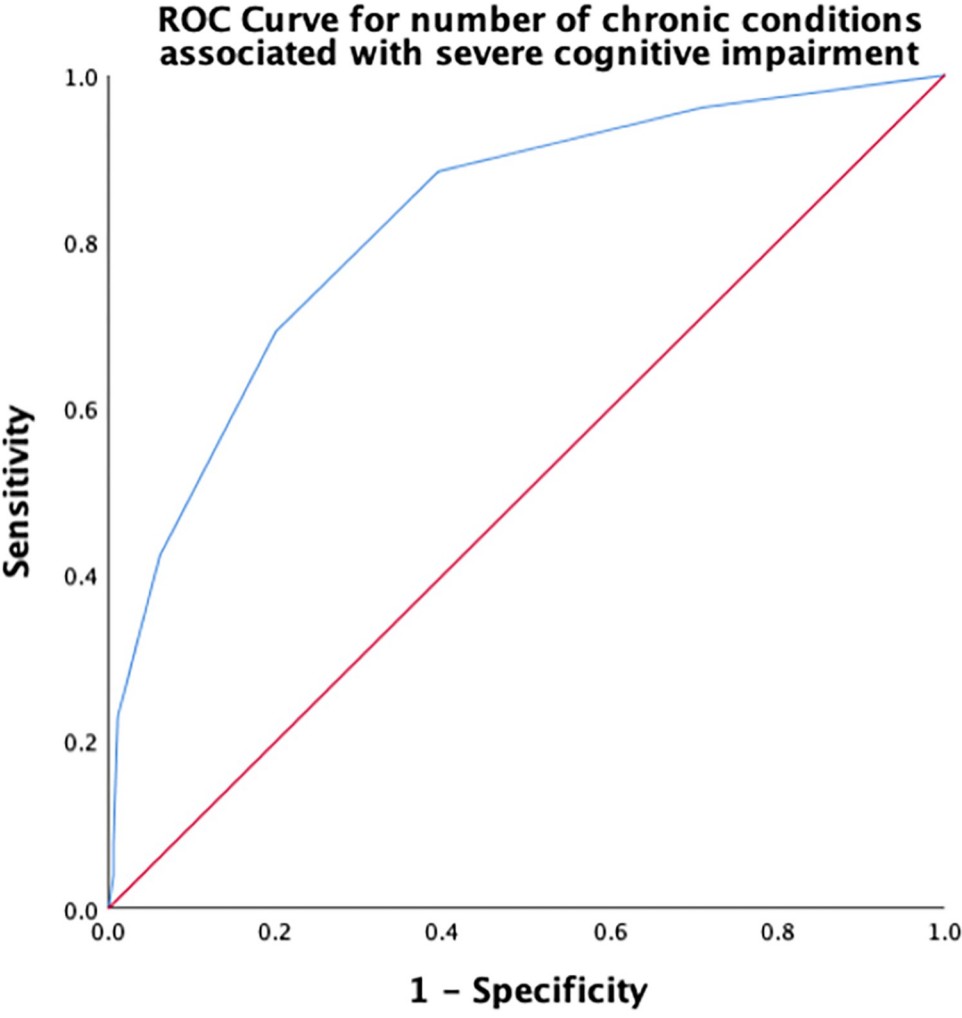

**Fig 1. ROC curves for the number of chronic conditions associated with severe cognitive impairment.**

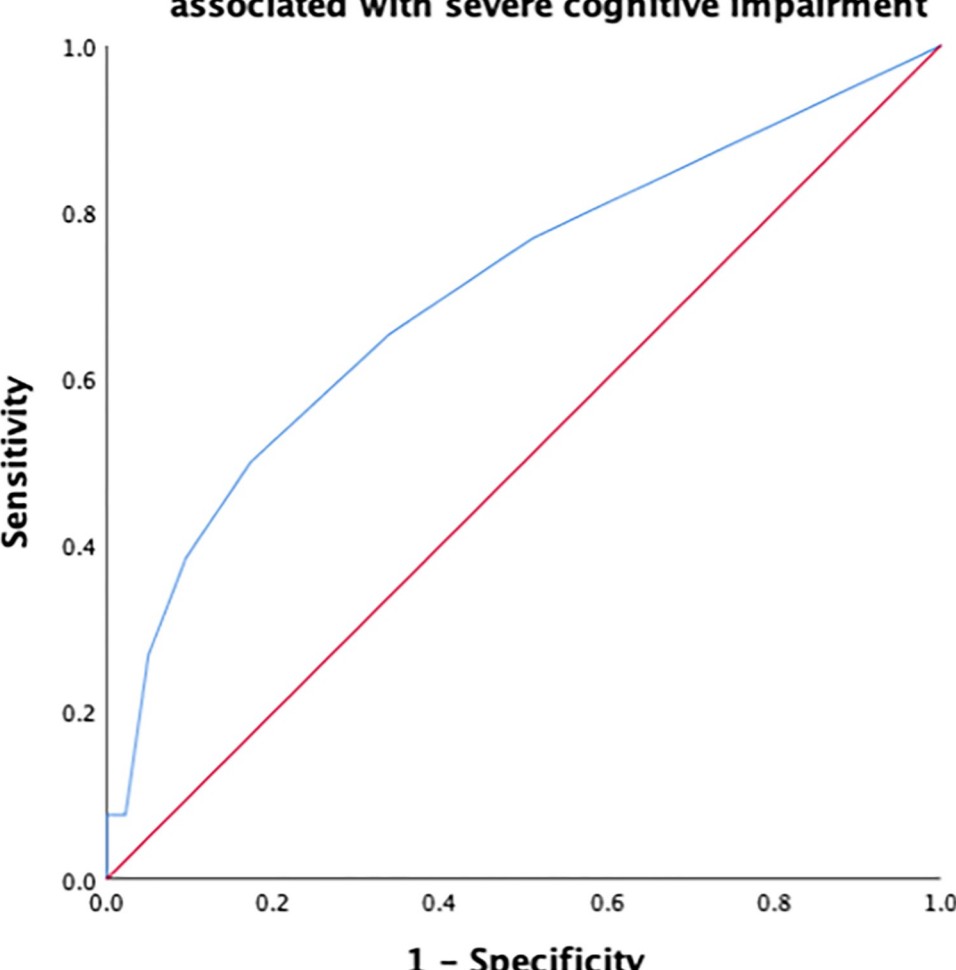

**Fig 2. ROC curves for the number of medications associated with severe cognitive impairment.**

score for PHQ-9 associated with severe cognitive impairment was ≥6.5 (Fig 3), whereas that for the physical function measures DGI (Fig 4), TUG (Fig 5), and 5XSST (Fig 6) associated with severe cognitive impairment were ≤ 21, ≥ 11.77 s, and ≥ 15.60 s, respectively.

The results for the sensitivity analysis showed similar results with the original dataset without imputation except for TUG (OR: 1.16, 95% CI [1.01, 1.34], p = 0.035) that became significant predictors for severe cognitive impairment.

## Discussion

This study determined the prevalence and associated risk factors of severe cognitive impairment in community-dwelling older adults in Saudi Arabia using the MoCA scale. The associated risk factors for severe cognitive impairment were the number of major chronic diseases, number of medications, and depressive symptoms. Other physical functions associated with severe cognitive impairment included decreased scores for the DGI and increased times for the TUG and 5XSST. These cut-off scores for the risk factors and physical functions

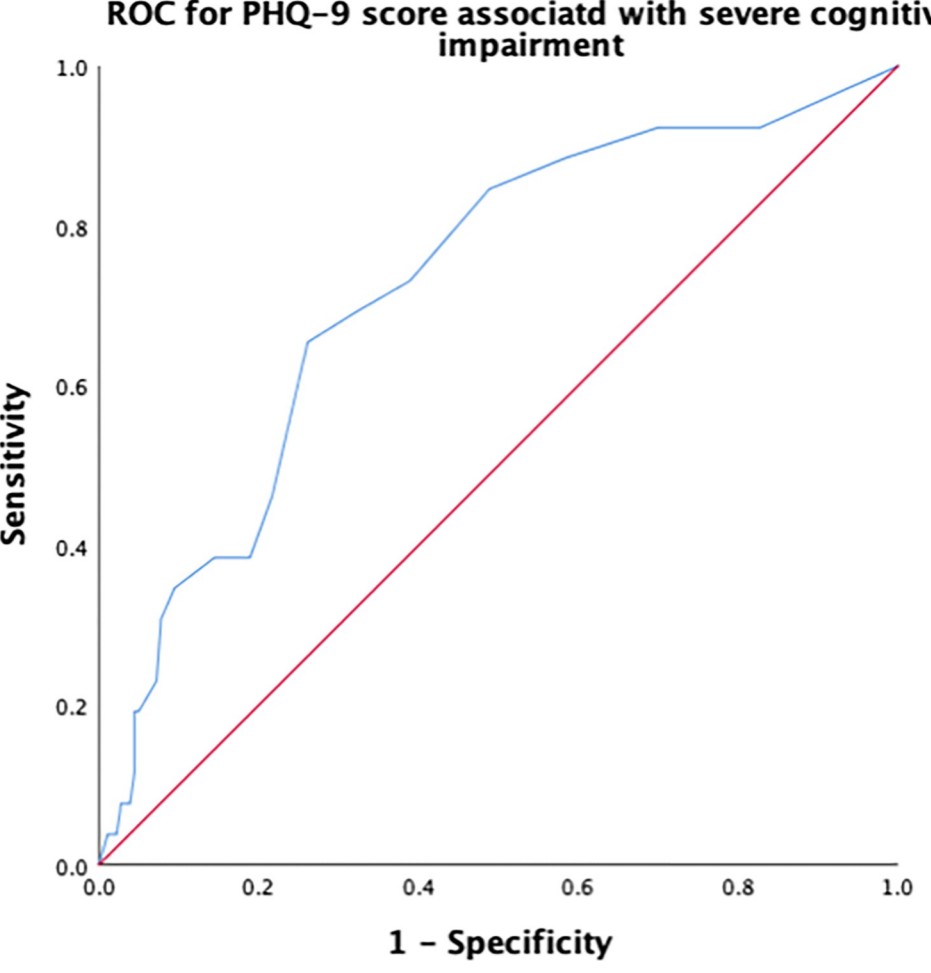

**Fig 3. ROC curves for the PHQ-9 score associated with severe cognitive impairment.**

measures of severe cognitive impairment in community-dwelling adults may be useful for future clinical practice.

Severe cognitive impairment was higher in women and the 60–64-year age group. These findings disagree with a recent study performed on older adults in Norway, which showed that women were associated with a higher score on MoCA evaluations [43]. However, another study investigating cognitive impairment among older adults in Greece concluded that age or gender does not affect the MoCA score [44]. In contrast, a recent meta-analysis and systematic review of 41 studies found that the prevalence of MCI in community-dwelling Chinese older adults increases with age [45]. These differences in cognitive impairment prevalence are likely due to diversity involving screening tools and diagnostic criteria.

The number of chronic conditions, number of medications, and depressive symptoms were associated with severe cognitive impairment in the current study. The association between health status, as measured by the presence of chronic diseases and cognitive function among older adults (n = 130) in Saudi Arabia, showed a non-significant correlation between the two [46]. Furthermore, a recent study in the US involving 17,899 older adults found that the presence of stroke, high blood pressure, diabetes, heart problems, and disease comorbidity were

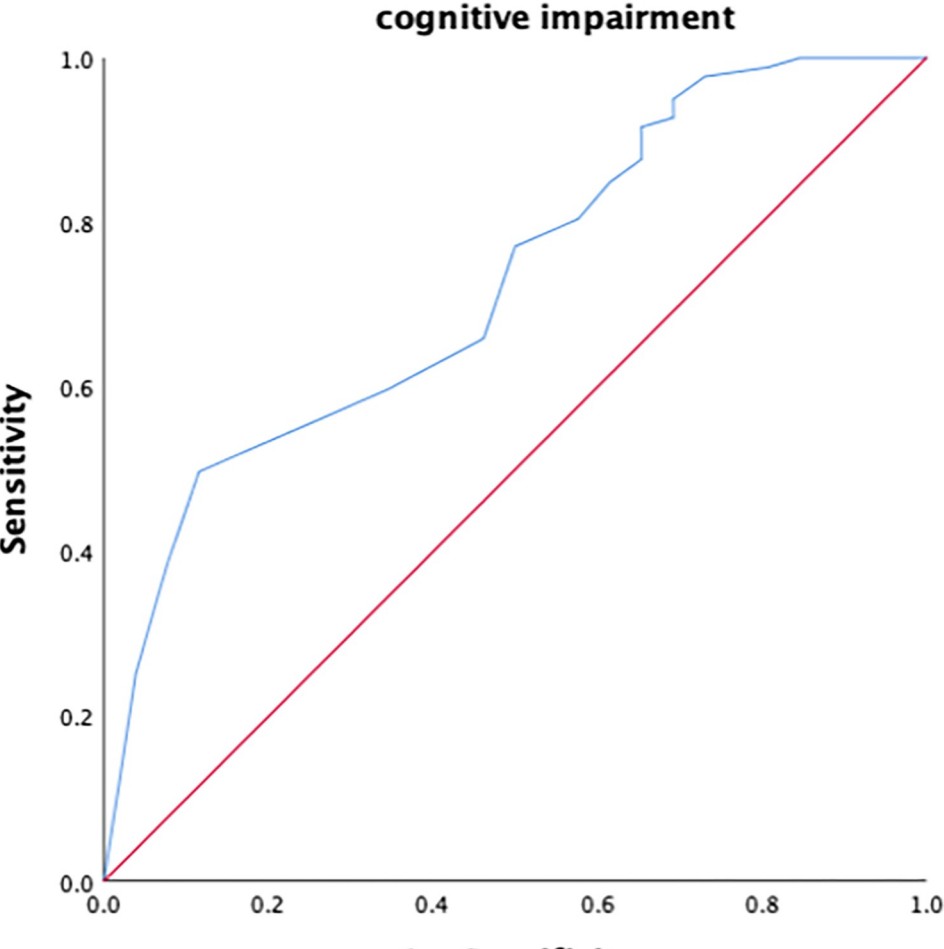

**Fig 4. ROC curves for the DGI score associated with severe cognitive impairment.**

associated with lower cognition levels [47]. They explained that the effects of chronic diseases on cognition level was partially mediated through depressive symptoms (19–54%) [47]. A national survey of 3,101 older adults reported that depressive symptoms, measured using PHQ-9, are associated with lower cognitive function [48]. Older adults who are at risk of depression are more likely to present with MCI [14]. The cause of depression (e.g. health-related diseases, social factors, or a combination of both) should be observed to explain the impact of chronic diseases and social factors on the development of depression and cognitive impairment.

Physical functions, including DGI, TUG, and 5XSST, exhibited an association with severe cognitive impairment, indicating a contribution to low cognitive function among older adults. Participants with severe cognitive impairment had lower DGI scores, and a cut-off score of ≤ 21 was associated with severe cognitive impairment. The relationship between cognitive functioning and gait performance, as measured using DGI, has been reported in a previous study, which revealed a strong relationship between cognition and gait performance in older

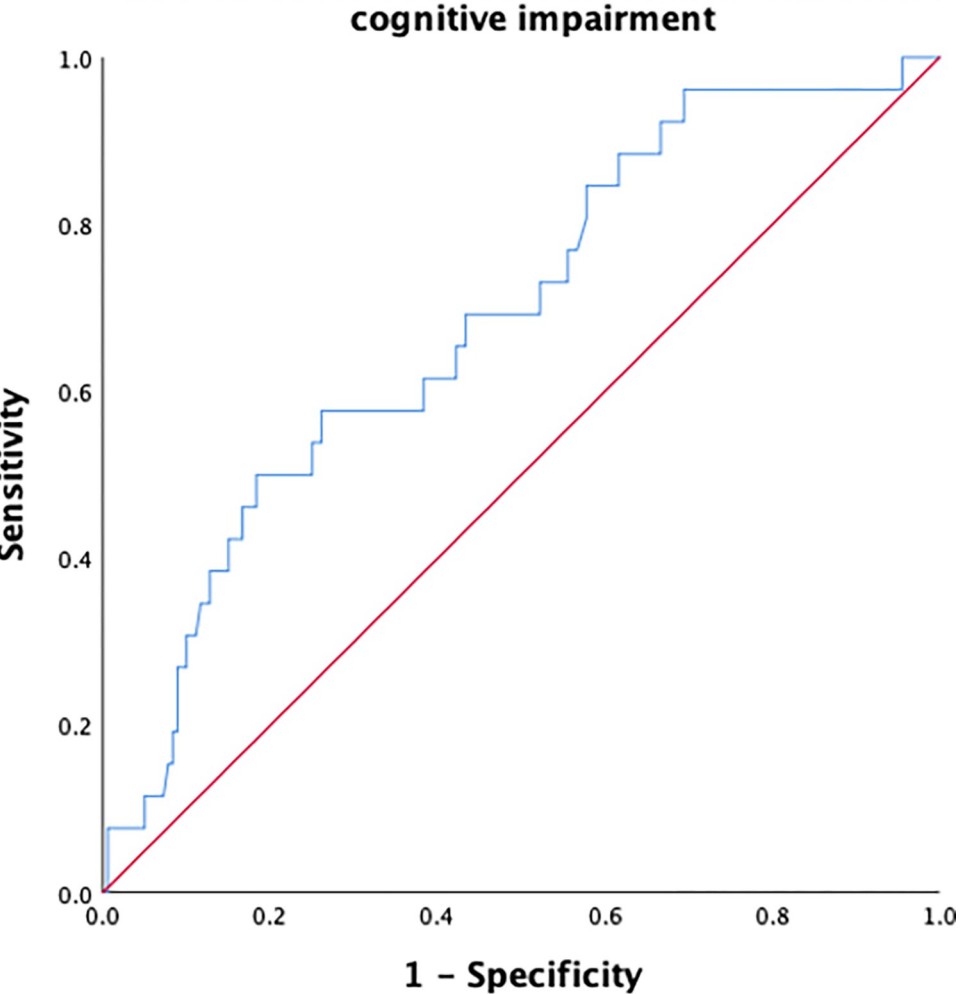

**Fig 5. ROC curves for the TUG score associated with severe cognitive impairment.**

adults ($>$ 65 years old) undergoing physical therapy interventions at home [17]. One possible explanation is that walking speed can be a sign of brain injury (white matter disease) associated with cognitive deterioration [16]. Cognitive functions are necessary for gait performance during ageing, with significant relationships between yaw amplitude during turns and attention or visual–spatial ability [15, 35]. In our cohort, participants with severe cognitive impairment took a considerable amount of time to complete the TUG tests. A cut-off score of $\geq$ 11.77 s in TUG performance was associated with severe cognitive impairment. A prolonged time to plan a movement during TUG test performance is associated with MCI in older adults [49]. This finding can be explained by the contribution of visual-spatial processing during turning to enable clear directional movement [15].

The time taken to complete the 5XSST was also higher in the severe cognitive impairment group. A cut-off score of $\geq$ 15.60 s was associated with severe cognitive impairment. This result is consistent with a previous study that included 7,421 community-dwelling older adults; 15 s in 5XSST was a sensitive value to predict moderate cognitive impairment [50]. A recent

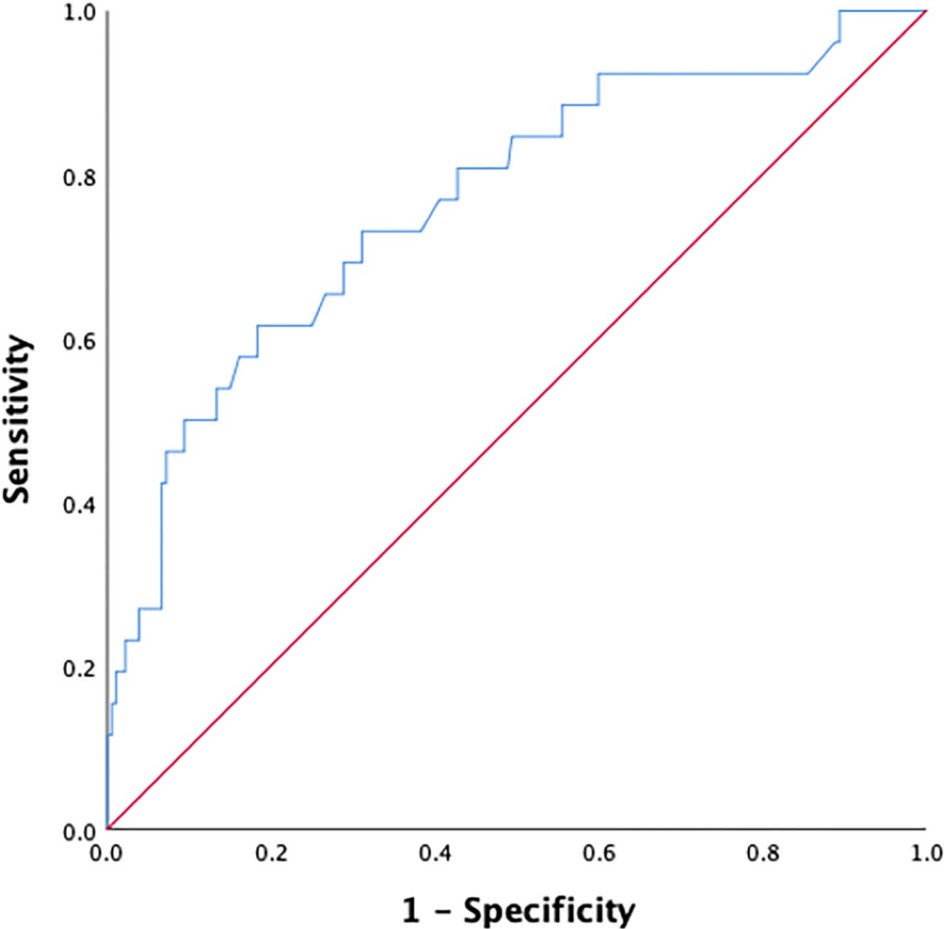

**Fig 6. ROC curves for the 5XSST score associated with severe cognitive impairment.**

logistic regression analysis showed that 5XSST is associated with cognitive functional decline in older adults [51]. Overall, the association between physical functions and cognitive impairment in our study reinforced the hypothesis that motor function is associated with cognitive function, as planning and decision-making are required for successful task completion.

In the current study, FRT and 6MWT were not associated with severe cognitive impairment. FRT was significantly associated with severe cognitive impairment in an unadjusted model with an odds ratio of 0.99. This indicated that a decreased in FRT distance increased the risk of severe cognitive impairment among the community-dwelling adults. However, after adjusting the model for potential covariates, this association did not reach a statistical significance, indicating a potential mediation of covariates involved in the association between FRT and severe cognitive impairment. 6MWT was not significantly associated with severe cognitive impairment in the community-dwelling adults, as evidenced using regression analysis. However, a significant difference was observed in the 6MWT between those with severe cognitive impairments and those with MCI or normal cognitive function using an

independent Student's *t*-test. This indicates potential confounding by other factors that were adjusted for in the regression model. Future research should examine this association with a larger sample size with consideration of other measures of mobility and balance.

We examined the cut-off scores for the number of chronic conditions using AUC and ROC. A cut-off score of approximately 2–3 number of chronic conditions was associated with severe cognitive impairment with an AUC of 82%. Although this AUC is considered to have excellent accuracy and discriminant ability ($0.8 \leq$ AUC $< 0.9$) [42]. These statistics provide valuable information (AUC = 82%) regarding the ability of our model to discriminate between cases and non-cases, and they are also commonly used in clinical and epidemiological research. Using ROC and AUC allowed us to examine the ability of each measure to discern severe cognitive impairment among the community-dwelling adults aged $\geq$50 years. However, the absence of a gold standard for the evaluation can limit the interpretability of the current results. AUC values for each functional measure may not reflect the true diagnostic accuracy for detecting severe cognitive impairment in this population. Therefore, the usefulness of these measures in clinical practice warrants future studies in similar populations. We believe that our findings may provide insights into the potential utility of functional measures, including the number of chronic diseases, number of medications, PHQ-9, DGI, TUG, and 5XSST, in the detection of severe cognitive impairment.

This study had several major strengths. First, this cross-sectional study employed a large sample of community-dwelling older adults in Saudi Arabia recruited through convenient sampling. The data collection focused on the risk factors. However, this study also had certain limitations, including lack of sufficient data to account for the underlying reasons for the factors involved and a limited number of participants from different regions. This study included only adults who were able to read and write in Arabic, which may have affected the representation of the elderly population because the illiteracy rate in Saudi Arabia for older adults is 59.29% [52]. Further research using larger sample sizes to examine and compare these findings in aged populations warranted. The study was cross-sectional; therefore, any significant association between cognitive impairment and risk factors cannot be definitively determined from the results, rather reverse causation stands as a possibility in the observational study design. Another limitation was the instruments used in the current study may have affected the validity and reliability of the results. We used a comprehensive battery of instruments to examine the factors associated, including functional performance, with severe cognitive impairment. Using a multidimensional approach may provide a more complete understanding of the relationship between these constructs and severe cognitive impairment. However, using multiple instruments may introduce measurement bias and impact the results regarding the prevalence of severe cognitive impairment in this population. To decrease the risk of measurement bias, we applied a rigorous instrument selection process based on established psychometric properties, reliability, validity, and clinical relevance, and ensured that all the instruments were administered and scored consistently across participants. Sensitivity analyses were conducted to assess the robustness of our findings against potential measurement errors. The results for imputations were similar to the results of the original dataset except for TUG that became a significant risk factor for severe cognitive impairment. This may indicate that an increase in the time for performing TUG was associated with severe cognitive impairment. It also indicates that the missing data affected the original results due to a lack of reaching significance level. In addition, using the self-reported data on chronic conditions, may be associated with recall bias. However, efforts were made to minimize bias as the data were collected using a standardized questionnaire and all the interviewers were well trained in mitigating information bias [23]. Objective measures for chronic conditions were not feasible in our study due to resource constraints and logistical limitations and recruitment feasibility from the community.

Thus, a rigorous data validation process was used to minimize the impact of the self-reported data on our findings. This process included assessing the robustness of the data via data cleaning, consistency checks, sensitivity analyses, and double validation from research assistants. Finally, this study used convenience sample, rather than random sampling. This can raise another form of selection bias that might affect the results. Future research should examine chronic diseases using physician diagnoses and diagnostic codes for health records. Another limitation was that the participants may have been fatigued due to the multiple assessments. Although efforts were made to divide the assessments between two sessions, the potential impact of fatigue should be considered in future research in order to minimize its impact on the test outcomes.

## Conclusion

The prevalence of severe cognitive impairment was moderate among community-dwelling adults in Saudi Arabia. Several risk factors were significantly associated with decline in cognitive function. Individuals with chronic conditions, a higher number of medications, and a score of $\geq 6.5$ in the PHQ-9 were more likely to have severe cognitive impairment. Deterioration in the DGI, TUG, and 5XSST test performance was associated with the development of severe cognitive impairment in community-dwelling adults. Further research is required to develop multimodal risk reduction approaches and examine their effectiveness in community-dwelling adults.

## Supporting information

**S1 Checklist. STROBE check list is available in the supporting information.**
(DOCX)

## Author Contributions

**Conceptualization:** Gamar R. Almutairi, Noura R. Almegbas, Rawan M. Alosaimi, Ibrahim A. Alfageh, Mohammed M. Alshehri, Shuruq F. Alanazi, Bader A. Alqahtani.

**Data curation:** Noura R. Almegbas, Maha A. Alqahtani, Shuruq F. Alanazi.

**Formal analysis:** Gamar R. Almutairi, Rawan M. Alosaimi, Bader A. Alqahtani.

**Methodology:** Norah A. Alhwoaimel.

**Supervision:** Ahmed S. Alhowimel, Aqeel M. Alenazi.

**Writing – original draft:** Ahmed S. Alhowimel, Norah A. Alhwoaimel, Aqeel M. Alenazi.

**Writing – review & editing:** Saleh G. Batook, Mohammed M. Alshehri, Norah A. Alhwoaimel, Aqeel M. Alenazi.

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
