## [Decision Letter · Decision Letter 0]

4 Apr 2024

PONE-D-24-07185Comorbidities, medications, depression, and physical performance measures are associated with severe cognitive impairments in community-dwelling adultsPLOS ONE

Dear Dr. Alenazi,

Thank you for submitting your manuscript to PLOS ONE. After careful consideration, we feel that it has merit but does not fully meet PLOS ONE’s publication criteria as it currently stands. Therefore, we invite you to submit a revised version of the manuscript that addresses the points raised during the review process.

We look forward to receiving your revised manuscript.

Kind regards,

Mohammad (Md) Jobair Khan, BSPT, MPH

Academic Editor

PLOS ONE

Reviewers' comments:

Reviewer's Responses to Questions

**Comments to the Author**

1. Is the manuscript technically sound, and do the data support the conclusions?

Reviewer #1: Yes

Reviewer #2: Partly

2. Has the statistical analysis been performed appropriately and rigorously? 

Reviewer #1: Yes

Reviewer #2: No

3. Have the authors made all data underlying the findings in their manuscript fully available?

Reviewer #1: Yes

Reviewer #2: Yes

4. Is the manuscript presented in an intelligible fashion and written in standard English?

Reviewer #1: No

Reviewer #2: Yes

5. Review Comments to the Author

Reviewer #1: The authors of this study have presented information concerning the associations between certain measurements and severe cognitive impairments in community-dwelling adults. Overall, the study is well-designed and drafted. However, there are opportunities for improvement in the draft before publication, as outlined in the comments and suggestions provided below.

1. General:

The language and grammar use need major enhancement.

2. Abstract:

A: Begin by formulating a concise main idea for your objective, followed by its elaboration. Consider incorporating one background sentence emphasizing the importance of aging and related impairments, akin to the approach taken in your introduction (it is clearly indicated in introduction).

B: Please consider refining the method to enhance clarity and ensure grammatical correctness.

3. Introduction:

A: Use simple past tense ('reached 2.4%') when referring to the previous statistic, as mentioned.

B: Depressive symptoms measurement was mentioned in the abstract, it wasn't specifically addressed in the introduction. Consider incorporating this aspect briefly into the introduction for clarity and coherence.

4. Discussion:

Insufficient discussion has been provided regarding the two physical measures, namely the Functional Reach Test and the 6-minute walk test. Please include this aspect in your discussion.

Reviewer #2: This manuscript reports on a cross-sectional study that examines the prevalence of severe cognitive impairment among community-dwelling older adults in Saudi Arabia and its association with comorbidities, medication use, depressive symptoms, and physical performance measures. The topic is of significant importance, aiming to fill a crucial gap in the literature by providing insights into factors associated with severe cognitive impairment. The attempt to conduct this initial stage of epidemiological investigation is commendable. However, to enhance the clarity and impact of the findings, there are several areas where improvements are recommended. Publication is advised after addressing the outlined major and minor issues.

Major issues:

1. Sampling Framework and Process:

The manuscript requires more detail regarding the sampling framework and process, including the number of individuals approached and the success rate. This information is vital for assessing the study's representativeness and potential selection bias.

2. Exclusion Criteria and Generalisability:

Given the exclusion criteria for the ability to read and write in Arabic, a discussion on the literacy rate among the elderly in Saudi Arabia is necessary. This will help evaluate the generalisability of the study results to the target population.

3. Data Collection and Chronic Disease Assessment:

The manuscript should clarify whether the clinical data collection included physical examinations or was solely based on verbal communication asking about chronic diseases. Furthermore, the approach to evaluating chronic diseases, given the variability in definitions and numbers in prior research, needs greater transparency. If the diagnosis is self-reported, the manuscript should address potential biases, such as recall and social desirability bias, and question the validity of referring to these as "clinical variables".

4. Participant Fatigue:

The potential for participant fatigue due to undergoing multiple assessments should be considered. The manuscript should describe any efforts made to prevent fatigue bias or discuss this potential bias in the discussion section if such measures were not taken.

5. Depressive Symptoms as Exposure:

The manuscript describes the measurement of depressive symptoms as part of the outcome measures, yet the study's aim suggests it as an exposure factor. This needs clarification for consistency and accuracy in the study's presentation.

6. Accuracy of Exposure Measures:

The manuscript's utilisation of self-reported data for comorbidities (not objectively measured) brings into question the accuracy of these exposure measures. This issue is crucial as it directly impacts the validity of the findings, including the reported odds ratios of severe cognitive impairment and risk factors. The reliance on ROC and AUC for inference, despite their potential to be significantly affected by biases, warrants a critical evaluation. It is imperative that the authors devote a substantial portion of the discussion to addressing and mitigating potential biases, including measurement and selection biases. This focus is essential to align the study with its objectives and enhance its relevance and credibility. An in-depth evaluation of these more fundamental aspects, such as the nature and impact of biases on the study's conclusions, would provide a more solid foundation for the research findings than the technical details provided by ROC and AUC analyses alone.

Minor issues:

1. Instrument Use and Measurement Bias:

The use of several instruments to measure outcomes raises questions about the study's inference regarding the prevalence of cognitive impairment and whether it may be affected by measurement bias. The authors should address the choice of instruments and their impact on study conclusions.

2. ROC and AUC Inference:

The use of ROC and AUC for several inferences, without a gold standard for evaluation, warrants a discussion on how this information contributes to future studies and the study's coherence.

The manuscript tackles a pivotal research area with significant implications for understanding cognitive impairment among the elderly. However, concerns regarding sampling, data collection, and potential biases necessitate careful consideration before publication. The study's strengths lie in its attempt to explore under-investigated associations within a specific population. Ensuring the clarity, consistency, and accuracy of the methodology and findings will greatly enhance the manuscript's contribution to the field.

6. PLOS authors have the option to publish the peer review history of their article (what does this mean?). If published, this will include your full peer review and any attached files.

Reviewer #1: No

Reviewer #2: No

---

## [Author Response · Author response to Decision Letter 0]

26 May 2024

Response to reviewers PLOS ONE

Dear Editor and reviewers, 

Thank you for your constructive feedback and comments that help us in improving our manuscript. All comments were considered with track changes and highlighted in yellow across the manuscript. 

Reviewer #1: The authors of this study have presented information concerning the associations between certain measurements and severe cognitive impairments in community-dwelling adults. Overall, the study is well-designed and drafted. However, there are opportunities for improvement in the draft before publication, as outlined in the comments and suggestions provided below.

1. General:

The language and grammar use need major enhancement.

RESPONSE: Thank you for your comment, we sent the manuscript for proofreading service (EDITAGE). We attached the certificate with the supplementary materials. 

2. Abstract:

A: Begin by formulating a concise main idea for your objective, followed by its elaboration. Consider incorporating one background sentence emphasizing the importance of aging and related impairments, akin to the approach taken in your introduction (it is clearly indicated in introduction).

RESPONSE: Thank you for your comment, we revised this accordingly. 

B: Please consider refining the method to enhance clarity and ensure grammatical correctness.

RESPONSE: Thank you for your comment, we sent the manuscript for proofreading service (EDITAGE). We attached the certificate with the supplementary materials. 

3. Introduction:

A: Use simple past tense ('reached 2.4%') when referring to the previous statistic, as mentioned.

RESPONSE: we revised the manuscript and sent it for a proofreading service. 

B: Depressive symptoms measurement was mentioned in the abstract, it wasn't specifically addressed in the introduction. Consider incorporating this aspect briefly into the introduction for clarity and coherence.

RESPONSE: Thank you for your comment, we added a paragraph related to depressive symptoms. 

4. Discussion:

Insufficient discussion has been provided regarding the two physical measures, namely the Functional Reach Test and the 6-minute walk test. Please include this aspect in your discussion.

RESPONSE: thank you for your comment, we added a paragraph in the discussion related to FRT and 6MWT. 

Reviewer #2: This manuscript reports on a cross-sectional study that examines the prevalence of severe cognitive impairment among community-dwelling older adults in Saudi Arabia and its association with comorbidities, medication use, depressive symptoms, and physical performance measures. The topic is of significant importance, aiming to fill a crucial gap in the literature by providing insights into factors associated with severe cognitive impairment. The attempt to conduct this initial stage of epidemiological investigation is commendable. However, to enhance the clarity and impact of the findings, there are several areas where improvements are recommended. Publication is advised after addressing the outlined major and minor issues.

RESPONSE: Thank you for your comments. We revised according to your suggestions. 

Major issues:

1. Sampling Framework and Process:

The manuscript requires more detail regarding the sampling framework and process, including the number of individuals approached and the success rate. This information is vital for assessing the study's representativeness and potential selection bias.

RESPONSE: Thank you for your comment, we revised and added more details to the methods section regarding the sampling method and recruitment. 

2. Exclusion Criteria and Generalisability:

Given the exclusion criteria for the ability to read and write in Arabic, a discussion on the literacy rate among the elderly in Saudi Arabia is necessary. This will help evaluate the generalisability of the study results to the target population.

RESPONSE: we included this in the limitation section with the percentage of illiteracy among older adults. 

3. Data Collection and Chronic Disease Assessment:

The manuscript should clarify whether the clinical data collection included physical examinations or was solely based on verbal communication asking about chronic diseases. Furthermore, the approach to evaluating chronic diseases, given the variability in definitions and numbers in prior research, needs greater transparency. If the diagnosis is self-reported, the manuscript should address potential biases, such as recall and social desirability bias, and question the validity of referring to these as "clinical variables".

RESPONSE: Thank you for your comment, we included the validity and reliability of self-reported chronic diseases to the methods, and we added this to the limitation section. 

4. Participant Fatigue:

The potential for participant fatigue due to undergoing multiple assessments should be considered. The manuscript should describe any efforts made to prevent fatigue bias or discuss this potential bias in the discussion section if such measures were not taken.

RESPONSE: Thank you for your comment and suggestion. We discussed this in the limitation section and future directions. 

5. Depressive Symptoms as Exposure:

The manuscript describes the measurement of depressive symptoms as part of the outcome measures, yet the study's aim suggests it as an exposure factor. This needs clarification for consistency and accuracy in the study's presentation.

RESPONSE: Thank you for your comment. We agree with you depression was considered as an exposure factor and we revised accordingly. 

6. Accuracy of Exposure Measures:

The manuscript's utilisation of self-reported data for comorbidities (not objectively measured) brings into question the accuracy of these exposure measures. This issue is crucial as it directly impacts the validity of the findings, including the reported odds ratios of severe cognitive impairment and risk factors. The reliance on ROC and AUC for inference, despite their potential to be significantly affected by biases, warrants a critical evaluation. It is imperative that the authors devote a substantial portion of the discussion to addressing and mitigating potential biases, including measurement and selection biases. This focus is essential to align the study with its objectives and enhance its relevance and credibility. An in-depth evaluation of these more fundamental aspects, such as the nature and impact of biases on the study's conclusions, would provide a more solid foundation for the research findings than the technical details provided by ROC and AUC analyses alone.

RESPONSE: thank you for bringing this up and we agree with you regarding the accuracy of exposure measures. We added a paragraph discussing the potential bias and the efforts made. We added this to the discussion section. 

Minor issues:

1. Instrument Use and Measurement Bias:

The use of several instruments to measure outcomes raises questions about the study's inference regarding the prevalence of cognitive impairment and whether it may be affected by measurement bias. The authors should address the choice of instruments and their impact on study conclusions.

RESPONSE: Thank you for your comment. We added a paragraph in the discussion section discussing the selection of instruments and measurement bias. 

2. ROC and AUC Inference:

The use of ROC and AUC for several inferences, without a gold standard for evaluation, warrants a discussion on how this information contributes to future studies and the study's coherence.

RESPONSE: Thank you for your comment. We added a paragraph in the discussion related to the concerns associated with ROC and AUC.

The manuscript tackles a pivotal research area with significant implications for understanding cognitive impairment among the elderly. However, concerns regarding sampling, data collection, and potential biases necessitate careful consideration before publication. The study's strengths lie in its attempt to explore under-investigated associations within a specific population. Ensuring the clarity, consistency, and accuracy of the methodology and findings will greatly enhance the manuscript's contribution to the field.

RESPONSE: thank you for your comment. We addressed all comments and suggestions in this manuscript to improve the manuscript.

---

## [Decision Letter · Decision Letter 1]

14 Jun 2024

PONE-D-24-07185R1Comorbidities, medications, depression, and physical performance measures are associated with severe cognitive impairments in community-dwelling adultsPLOS ONE

Dear Dr. Alenazi,

Thank you for submitting your manuscript to PLOS ONE. After careful consideration, we feel that it has merit but does not fully meet PLOS ONE’s publication criteria as it currently stands. Therefore, we invite you to submit a revised version of the manuscript that addresses the points raised during the review process.

**Thank you for your effort to improve the manuscript. The reviewer additionally made minor comments. Please revise the manuscript following the comments.**

We look forward to receiving your revised manuscript.

Kind regards,

Ryota Sakurai, Ph.D.

Academic Editor

PLOS ONE

Journal Requirements:

Reviewers' comments:

Reviewer's Responses to Questions

**Comments to the Author**

1. If the authors have adequately addressed your comments raised in a previous round of review and you feel that this manuscript is now acceptable for publication, you may indicate that here to bypass the “Comments to the Author” section, enter your conflict of interest statement in the “Confidential to Editor” section, and submit your "Accept" recommendation.

Reviewer #1: All comments have been addressed

Reviewer #2: (No Response)

2. Is the manuscript technically sound, and do the data support the conclusions?

Reviewer #1: (No Response)

Reviewer #2: Yes

3. Has the statistical analysis been performed appropriately and rigorously? 

Reviewer #1: Yes

Reviewer #2: Yes

4. Have the authors made all data underlying the findings in their manuscript fully available?

Reviewer #1: Yes

Reviewer #2: Yes

5. Is the manuscript presented in an intelligible fashion and written in standard English?

Reviewer #1: Yes

Reviewer #2: Yes

6. Review Comments to the Author

**Reviewer #1: **(No Response)

**Reviewer #2:** I can see much improvement from the previous manuscript. Here are additional comments for the authors to consider:

BMI is not demographic data, but it is listed under the heading of demographics. I suggest changing the headline title to reflect the content better.

Marital status was mentioned in the method section but it was not included in Table 1. Please include this information for completeness.

In the introduction, the authors stated that the study aimed to examine the prevalence of severe cognitive impairment and its associated factors, later referred to as exposures. Since this is a cross-sectional study, these factors may be subject to reverse causality. It is crucial to acknowledge that the relationships identified could be due to reverse causality and should not be used definitively to determine the risk of severe cognitive impairment.

Please clarify if there is any specific reason behind the variable choices in including the adjusted model.

There is a missing period at line 227. Please correct this punctuation error.

In line 264, the steps taken to decrease measurement bias are not clearly stated. The author later describes the data validation process in lines 303 to 305, which includes data cleaning, consistency checks, sensitivity analysis, and double validation from the research assistant. In lines 363 to 365, the author discusses efforts to minimise information bias. To maintain consistency and improve the flow of the text, consider reorganising the discussion section, particularly the parts that address the limitation of measurement bias and the efforts made in the study.

In line 267, the authors mentioned conducting sensitivity analysis. Could you specify what type of sensitivity analysis was conducted? If possible, include this information in the supplemental analysis.

The authors mentioned subgroup results in the discussion section at line 269 (Severe cognitive impairment was higher in women and the 60–64-year age group.) However, the result was not confirmed in any tables or the results section. Please ensure all findings are included and clearly presented.

7. PLOS authors have the option to publish the peer review history of their article (what does this mean?). If published, this will include your full peer review and any attached files.

Reviewer #1: No

Reviewer #2: No

---

## [Author Response · Author response to Decision Letter 1]

22 Jul 2024

Response to reviewers

Dear Editor and reviewers, thank you for your valuable feedback on our manuscript that helps us in improving the paper. We responded to each comment and provide responses with boldface across the manuscript. 

Reviewer #2: I can see much improvement from the previous manuscript. Here are additional comments for the authors to consider:

1. Comment: BMI is not demographic data, but it is listed under the heading of demographics. I suggest changing the headline title to reflect the content better.

Response: ‘anthropometric’ has been added to the headline title and table 1 

2. Comment: Marital status was mentioned in the method section, but it was not included in Table 1. Please include this information for completeness.

Response: the following sentence has been added to table 1

Marital status was dichotomized as married or divorced/widowed. We also included this in the table1. 

3. Comment: In the introduction, the authors stated that the study aimed to examine the prevalence of severe cognitive impairment and its associated factors, later referred to as exposures. Since this is a cross-sectional study, these factors may be subject to reverse causality. It is crucial to acknowledge that the relationships identified could be due to reverse causality and should not be used definitively to determine the risk of severe cognitive impairment.

Response: The following sentence has been added to the limitation of study: 

The study was cross-sectional; therefore, any significant association between cognitive impairment and risk factors cannot be definitively determined from the results, rather reverse causation stands as a possibility in the observational study design.

4. Comment: Please clarify if there is any specific reason behind the variable choices in including the adjusted model. 

5. Response: Thank you for your comment. The variables included in the adjusted model were selected based on their relevance in the literature and their potential association with cognitive impairment as mentioned in the introduction (line 88-99). The adjusted model for age, gender and BMI aimed to control for key factors that have been previously shown to influence cognitive outcome. To clarify this point, the second paragraph of the statistical analysis section has been amended to read as following: 

Multiple binary logistic regression, odds ratio (OR), and 95% confidence interval (95% CI) were used to assess the potential associations between the risk factors and physical function measures of severe cognitive impairment. The variables included in the regression model were selected based on their relevance and potential association with cognitive impairment. The following two models were developed based on adjustments: model 1 was unadjusted, and model 2 was adjusted for age, sex, and BMI to control the key factors that have been shown to influence cognitive outcome. Previous studies employed unadjusted models [10, 41]. Missing variables were handled using case-wise deletion.

6. Comment: There is a missing period at line 227. Please correct this punctuation error.

Response: Corrected 

7. Comment: In line 264, the steps taken to decrease measurement bias are not clearly stated. The author later describes the data validation process in lines 303 to 305, which includes data cleaning, consistency checks, sensitivity analysis, and double validation from the research assistant. In lines 363 to 365, the author discusses efforts to minimise information bias. To maintain consistency and improve the flow of the text, consider reorganising the discussion section, particularly the parts that address the limitation of measurement bias and the efforts made in the study.

Response: The discussion section reorganized.

8. Comment: In line 267, the authors mentioned conducting sensitivity analysis. Could you specify what type of sensitivity analysis was conducted? If possible, include this information in the supplemental analysis.

Response: Thank you for your comment. The dataset has some missing observations for some of the key predictor variables. We performed multiple imputations and compared the results. The results for the sensitivity analysis showed similar results with the original dataset without imputation except for TUG (OR: 1.16, 95% CI [1.01, 1.34], p=0.035) that became significant predictors for severe cognitive impairment. We added the sensitivity analysis to the methods and its results to the results section for clarification. We also added to the discussion this clarification. 

9. Comment: The authors mentioned subgroup results in the discussion section at line 269 (Severe cognitive impairment was higher in women and the 60–64-year age group.) However, the result was not confirmed in any tables or the results section. Please ensure all findings are included and clearly presented.

Response: This point mentioned in the results: 

“The prevalence of severe cognitive impairment among community-dwelling adults was 12.6% (n = 26); this was higher in women than that in men and in the 60–64-year age compared with that in the other age categories” 

A clarification of the number of men/women has been added to table 1.

---

## [Editor Report · Decision Letter 2]

19 Aug 2024

Comorbidities, medications, depression, and physical performance measures are associated with severe cognitive impairments in community-dwelling adults

PONE-D-24-07185R2

Dear Dr. Alenazi,

We’re pleased to inform you that your manuscript has been judged scientifically suitable for publication and will be formally accepted for publication once it meets all outstanding technical requirements.

Kind regards,

Ryota Sakurai, Ph.D.

Academic Editor

PLOS ONE

Additional Editor Comments (optional):

Thank you for your effort in improving the manuscript. The manuscript has been amended appropriately based on the reviewers' comments.
---

## [Editor Report · Acceptance letter]

10 Sep 2024

PONE-D-24-07185R2 

PLOS ONE

Dear Dr. Alenazi, 

I'm pleased to inform you that your manuscript has been deemed suitable for publication in PLOS ONE. Congratulations! Your manuscript is now being handed over to our production team.

Kind regards, 

on behalf of

Dr. Ryota Sakurai 

Academic Editor

PLOS ONE